# Motor Influence in Developing Auditory Spatial Cognition in Hemiplegic Children with and without Visual Field Disorder

**DOI:** 10.3390/children9071055

**Published:** 2022-07-15

**Authors:** Elena Aggius-Vella, Monica Gori, Claudio Campus, Stefania Petri, Francesca Tinelli

**Affiliations:** 1Unit for Visually Impaired People (U-VIP), Center for Human Technologies, Istituto Italiano di Tecnologia, 16152 Genoa, Italy; elenaaggiusvella@gmail.com (E.A.-V.); monica.gori@iit.it (M.G.); claudio.campus@iit.it (C.C.); 2Institute for Mind, Brain and Technology, Ivcher School of Psychology, Inter-Disciplinary Center (IDC), Herzeliya P.O. Box 11000, Israel; 3Department of Developmental Neuroscience, Laboratory of Vision, IRCCS Fondazione Stella Maris, 56128 Pisa, Italy; stefania.petri@fsm.unipi.it

**Keywords:** auditory, visual, space, motor, children, hemiplegia

## Abstract

Spatial representation is a crucial skill for everyday interaction with the environment. Different factors seem to influence spatial perception, such as body movements and vision. However, it is still unknown if motor impairment affects the building of simple spatial perception. To investigate this point, we tested hemiplegic children with (HV) and without visual field (H) disorders in an auditory and visual-spatial localization and pitch discrimination task. Fifteen hemiplegic children (nine H and six HV) and twenty with typical development took part in the experiment. The tasks consisted in listening to a sound coming from a series of speakers positioned at the front or back of the subject. In one condition, subjects were asked to discriminate the pitch, while in the other, subjects had to localize the position of the sound. We also replicated the spatial task in a visual modality. Both groups of hemiplegic children performed worse in the auditory spatial localization task compared with the control, while no difference was found in the pitch discrimination task. For the visual-spatial localization task, only HV children differed from the two other groups. These results suggest that movement is important for the development of auditory spatial representation.

## 1. Introduction

How we develop a spatial representation of the environment is a topic that has been widely studied over the last decades [1,2,3]. Neuropsychological studies have shown that space around us is subdivided into several portions based on different anatomical and neural networks. Indeed, studies on patients affected by hemispatial neglect [4,5,6,7], electrophysiological works [8], and studies on peripersonal space [9,10,11,12] show that our brain does not represent space as a unitary dimension but that space representation is split up into different portions concerning the body position, i.e., near and far space [13], frontal and rear space [6,7,14,15], space around specific parts of the body [8,12], and space above and below the head in the frontal field [16].

Auditory spatial localization is very important for human effectiveness and personal safety. The sound of a weapon, vehicle, or an approaching person can usually be heard much earlier than the source of the sound can be seen. Knowing where to listen improves situational awareness, speech perception, and sound source identification in the presence of other sound sources (e.g., [17,18]).

There is a consensus about the crucial role of visual experience in calibrating auditory spatial skills and in guiding the maturation of spatial cognition, more in general [19,20]. Vision has, in fact, advantages over the other senses in encoding spatial information because it ensures the simultaneous perception of multiple stimuli in the environment [21].

However, as supported by extensive line of research [8,12,13], our brain elaborates spatial representation based on the possibility to act directly on it (within/outside hand-reaching distance), leading movement to be fundamental in spatial skills. The importance of movement in spatial cognition is well explained by the motor-oriented approach, which assumes that spatial relationships are coded by body movement in space [22]. Moreover, the representation of space is modulated by actions: training with a tool that modifies the representation of the body (i.e., the tool becomes perceived as a part of the body, embodied cognition) can extend the size of peripersonal space, leading to perceive what was previously far away as to be closer [23]. In agreement with this idea, animal studies have also shown that space representation is the secondary result of movement. This means that space coding is the result of the construction of multiple space representations that may be related to a specific class of actions. This suggests that the concept of one single parietal center for space perception is no longer sustainable [24]. It has been shown that space coded by area F4 in animals is based on an egocentric body part-centered frame of reference. The VIP–F4 circuit (involved in encoding peripersonal space) seems to process the spatial location of a stimulus according to a body part-centered frame of reference and transforms object locations into appropriate movements toward them. Based on this evidence, it is possible to think that a motor deficit to one side of the body can produce an impairment in locating stimuli in the space.

In this study, we hypothesize that if motor abilities are crucial for space representation, then unbalanced motor competencies between the two sides of the body, such as in hemiplegic subjects, should affect the perception of space. Indeed, it might be associated with an impaired body representation that, in turn, could make it impossible to use an egocentric reference frame to solve spatial tasks correctly. If our hypothesis is correct, we may expect to observe altered spatial processing in hemiplegic children. To test our hypothesis, we investigated audio and visual-spatial representation in hemiplegic children. Since the visual modality can solve the spatial task without the necessity to refer to the body as a reference, we expect that this deficit should be less evident in the visual modality. Indeed, vision can solve spatial tasks through an allocentric reference frame [3]. Children and adolescents with congenital hemiplegia with or without visual field disorder and typical children were requested to perform two different audio tasks (auditory spatial and pitch discrimination task) in the frontal and back zone (where movement is naturally not possible) and a visual task (visual-spatial discrimination task) in the frontal zone giving a verbal response. Firstly, we investigated how front and back auditory spaces were perceived in children with typical development and in children with hemiplegia and how much motor impairment can play a role in auditory space perception; secondarily, we evaluated the weight that a visual disorder can play in the auditory perception of space in children with congenital hemiplegia.

## 2. Materials and Methods

### 2.1. Patients

Fifteen hemiplegic children (Mean age 8.9, SD 0.5, 9 females), 5 with left hemiplegia and 10 with right hemiplegia, and 20 children with typical development (Mean age: 9.6, SD 0.3, 10 females) took part in the experiment. The hemiplegic group was split into two subgroups: 9 hemiplegic children without visual field defects and 6 hemiplegic children also presenting a visual deficit (see Table 1 for details). Patient enrollment is well described in the following flowchart below (Figure 1)

Inclusion criteria for hemiplegic children were: (i) congenital brain lesion; (ii) intelligence quotient in the normal range or in the border area; (iii) a behavioral or computerized evaluation of the visual field, (iv) no history of hearing impairment (all children underwent an audiometric examination that was found to be normal in the first few months of life); (iv) no auditory neglect and (v) normal or corrected-to-normal vision and absence of peripheral visual deficits.

Exclusion criteria for hemiplegic children were the presence of bilateral lesions or more global brain damage (e.g., meningitis, hypoxic-ischemic encephalopathy).

Exclusion criteria for typical children included any history of problems that may have resulted in neural, sensorial, or cognitive dysfunction.

All children who took part in the experiment were evaluated at the IRCCS Stella Maris Scientific Institute (hemiplegic children) and at the Italian Institute of Technology (typical children).

All parents of each participant gave written informed consent in accordance with the Declaration of Helsinki, and the study was approved by the ethics committee of the local health service (Comitato Etico, ASL3 Genovese, Italy) and by the Ethics Committee of Meyer’s Hospital (n. 298/2021 IIT_UVIP_MySpace).

### 2.2. Setup and Stimuli

Participants were seated in the center of an array of 11 loudspeakers and red LEDs positioned in an arc on a table. The center of each loudspeaker and LED was positioned at 7 cm from the center of the next. The position of the speaker array was fixed in the approximate center of the room. The seating distance was 45 cm from the middle loudspeaker; the participant’s ears were aligned with the loudspeakers so that sounds were presented at 0° elevation (Figure 2).

Participants performed an auditory spatial localization task on the horizontal plane and a pitch discrimination task. This last task was used as a control to be sure that sounds were equally perceived in every space by all subjects and that sound discrimination was intact. In order to understand the role of vision in basic spatial cognition, all healthy children and 13 hemiplegic children (4 with visual field disorder and 9 without visual field disorder) performed an additional control task: the visual-spatial localization task. Two subjects refused to do this task.

Auditory spatial discrimination and pitch discrimination tasks were carried out in two different orientations with respect to the participant’s body position: frontally (so that the midpoint of the loudspeaker array was at 0° azimuth relative to the participants) and from the back (midpoint of the loudspeaker array at 180° azimuth). Auditory stimuli were 300 and 800 kHz burst sounds, randomized between trials, both with a duration of 100 ms. Sounds were generated, and responses were recorded using a custom-written MATLAB (Mathworks) code.

Auditory spatial discrimination and pitch discrimination tasks were based on the same procedures and stimulation but differed for the task question: in the spatial discrimination task, children were required to discriminate the spatial position of the sound with respect to their body without taking into account the different pitch. In pitch discrimination, however, children were required to discriminate between pitches (high or low) without considering their spatial position.

Visual-spatial discrimination was identical to auditory spatial discrimination, but instead of sounds, it used red lights and was performed only in the frontal space. Children were required to discriminate the spatial position of the light with respect to their bodies.

### 2.3. Tasks and Procedures

The experimental room was a quiet, anechoic room. For the auditory tasks, participants were blindfolded and instructed that they would hear sounds originating from loudspeakers positioned around them.

For the auditory spatial localization task, participants heard one sound in each trial. The sound could be presented on any loudspeaker. Participants reported verbally whether the stimulus was to the right or left with respect to their body, without considering the 2 different pitches. The response was recorded by the experimenter using the response interface. The same procedure was used with visual stimuli.

In the pitch discrimination task, the same 2 auditory stimuli were used, and subjects were asked to discriminate if the sound was at 300 (low) or 800 hrz (high) without taking into consideration the spatial position of the sound.

No feedback was given, and the response time was not constrained. Each participant performed 66 trials for the auditory blocks (33 trials for each pitch and each side in a randomized way) and 36 trials for the visual block (18 for each side). Data collection lasted approximately 1 h. For the spatial discrimination task, we calculated the probability that the response to the sound or light was to the right. The data sets were fitted with cumulative Gaussian functions (see Figure 3). Figure 3 shows an example of a psychometric function from a typical participant for the spatial bisection task. The probability that the response to the sound or light was to the right is plotted as a function of sound or light position. For each participant and condition, the standard deviation (σ) of the fit, which provides an estimate of the slope of the psychometric function, was taken as the estimate of threshold/precision (JND). The midpoint of the function is represented by the Point of Subjective Equality (PSE). For the pitch discrimination task, we calculated the proportion of correct responses.

### 2.4. Analysis

Firstly, we investigated if portions of space were differently represented in typical and hemiplegic children. To this end, after angle normalization, we performed a repeated measure ANOVA on the mean thresholds of the auditory spatial task, which factored space (front and back) and groups (T = typical, H = hemiplegic children without visual field disorder, and HV = hemiplegic children with visual field disorder).

In the second analysis, we evaluated the performance of the pitch discrimination task by comparing the three groups in the 2 spaces. A repeated measure ANOVA was performed on the accuracy, factoring spatial region (front, back) and groups (T, H, HV). Post hoc comparisons were conducted using t-tests, and *p* < 0.05 was considered significant after applying the Bonferroni correction for multiple comparisons.

In a third analysis, we performed a repeated measure ANOVA on the mean thresholds of the visual-spatial task using the groups (T, H, and HV).

The possible correlation between front auditory spatial discrimination and MACS values in the whole group of hemiplegic children was studied by Spearman’s test (*p* < 0.05 was considered significant).

## 3. Results

### 3.1. Auditory Spatial Discrimination Task

The results of the first analysis showed a significant interaction between spatial regions and groups (*F*_(2,32)_ = 4.4, *p* = 0.02, generalized eta squared (ges) = 0.05). The auditory spatial discrimination ANOVA also showed a main group effect (F_(2,32) =_ 14, *p* < 0.01, ges = 0.4). Typical children performed better than H-children (t_(27)_ = 5.9, *p* < 0.01) and HV-children (t_(24)_ = 3.1, *p* = 0.01), while no difference was found between the H and HV-groups (t_(13)_ = 1.2, *p* > 0.05). Moreover, a main effect of spatial region (*F*_(1, 32)_ = 21, *p* < 0.01, generalized eta squared (ges) = 0.1), showing a better performance in the frontal space compared to the back (*t*_(34)_ = −4.2, *p* < 0.01), was found.

In the frontal space, T-children performed better than H-children (t_(27)_ = 7.7, *p* < 0.01) and HV-children (t_(24)=_ 2.6, *p* = 0.04), while no difference between the two hemiplegic groups (H and HV) was found (t_(13)_= 8.4, *p* > 0.05) (see Figure 4a). The same results were found in the back (Figure 4b): the T-group performed better than the H-group (t_(27)_ = 6.9, *p* < 0.01) and HV-group (t_(24)_ = 2.9, *p* = 0.02), while no difference was found between the two hemiplegic groups in both spaces (*p* > 0.05).

As expected, a paired t-test (Figure 4), showed that the group of typical children (Figure 5a) performed better in the frontal space, compared to the back (t_(19)_ = −3.7, *p* < 0.01); a similar result was found in the H-children group (t_(8)_ = −3, *p* = 0.04) (Figure 5b), while no difference was found in the HV-group (t_(5)_ = −1.6, *p* > 0.05) (Figure 5c).

In each group, we calculated the ratio between front and back performance (Figure 6). Results showed no differences between groups (all *p* > 0.05).

### 3.2. Pitch Discrimination Task

The accuracy ANOVA (Figure 7) which factored spatial position (front vs back) and groups (T, H and HV groups) showed no differences between spaces (F_(1,31)_ = 2.8, *p* > 0.05), groups (F_(2,31)_ = 3.2, *p* > 0.05) and no interaction between spaces and groups (F_(2,31)_ = 1.6, *p* > 0.05).

### 3.3. Visual-Spatial Discrimination

This task was used as a control task to investigate if the role of movement on spatial representation was confined to the auditory space or was generalizable to the visual-spatial task. The ANOVA showed a difference between groups (F_(2,30)_ =5.6, *p* < 0.01). The T-group performed better than the HV-group (t_(22)_ = 3.4, *p* = 0.007), while no difference was found between the two hemiplegic groups (t_(11)_ = −1.7, *p* > 0.05) and between the H-group and T-group (t_(27)_ = 0.9, *p* > 0.05) (Figure 8).

### 3.4. Correlation between MACS Values and JND at Front Auditory Spatial Discrimination Task

No significant correlation was found between the MACS values and the front auditory spatial discrimination task in the whole group of hemiplegic children (H + HV) (*p* = 0.721).

## 4. Discussion

It is well demonstrated that body movements [8,25,26,27] and vision [3,16,28,29] influence spatial cognition [30]. Though many studies have been published about the influence of visual disorders on auditory spatial perception [31,32], less is known about if and how movement disorders affect auditory spatial cognition.

What is clear, at the moment, is that the motor and auditory systems are strictly related to the brain. Neuroimaging studies have shown, in fact, significant activation in the supplementary motor area, pre-supplementary motor area, inferior frontal gyrus, middle frontal gyrus, and cerebellum related to rhythm processing, such as passive listening to rhythm and rhythm production [33]. The tendency of auditory rhythms to make us move has been widely demonstrated in the past years, so that has also received interest in clinical contexts, as it can be used to simulate and modulate the motor system of patients with movement disorders (such as Parkinson disease) simply by presenting auditory rhythms [34,35,36,37,38]. On the contrary, less is known about the effect of motor impairment on the auditory system and spatial cognition, in particular in children. Some months ago, Martin and Trauner [39] reported auditory neglect in children with early unilateral brain damage from perinatal stroke. The authors found that children who had experienced left hemisphere perinatal strokes were significantly better at localizing sounds on the left side of space than on the right side of space and that response times improved with age on a normal trajectory relative to controls in the left hemispace. In contrast, they did not normally improve in the right hemispace. Children with right hemisphere perinatal strokes were significantly worse at localizing sounds on the right side of space relative to typically developing controls and did not follow control trajectories for improvement in response times on the left or the right sides of space. It means that left hemisphere perinatal strokes may result in contralateral auditory neglect, while right hemisphere perinatal strokes may result in bilateral auditory neglect.

In that study, however, the authors were interested in the correlation between spatial auditory impairment and brain lesions (in particular cortical parietal brain lesions, where the sound is processed), so they did not analyze the correlation with motor impairment (also because it was present only in 13 subjects) or with visual field disorder.

Our study is, to our knowledge, the first one focused on the role of movement in developing sensory spatial skills in hemiplegic children. For this reason, we enrolled only children with congenital brain lesions and with motor impairment (hemiplegia), paying specific attention to subjects with or without visual field disorders. The enrolled group, keeping into account all the inclusion criteria, was limited to fifteen hemiplegic children: five with left hemiplegia and ten with right hemiplegia, so it was not possible to do specific analysis according to the side of the lesion. We tested hemiplegic children as a reference to judge space, as this kind of motor impairment affects body representation, leading to an impossibility to use the body. To test this hypothesis, we administered three tasks: in the first two, subjects had to localize sound with respect to their body position in the auditory and visual modality. In the last task, used as a control task, subjects had to recognize if the sound was a high or low pitch while not considering from which part of space it was produced. As expected, in both spaces, we found worse performance in the auditory spatial localization task in children with hemiplegia (both without and with visual field disorder) compared to children with typical development, while no difference between the two hemiplegic groups was found in both spaces. Importantly, no differences were reported between spaces among the three groups in the pitch discrimination task. This suggests that motor impairment influences selectively spatial representation since that auditory perceptual skills were similar across spaces and groups. This is in agreement with the literature on blind children, who develop good spatial hearing abilities only according to the richness of perceptual experiences in life. Studies on young blind children demonstrated, in fact, an inability to identify the position of sonorous objects embedded in space before 12 months [40] (while sighted children start around 5 months [41]) and a compromised ability to represent the relation of sounds in space both in the horizontal [31] and sagittal planes [28]. The lack of vision causes a delay in the development of mobility and locomotor skills [42], which in turn causes visually impaired children to accumulate much less spatial experience compared to their sighted peers [43].

More interestingly, we specified that the three groups adopt the same strategy/cognitive process based on a body-centered coordinate system. Indeed, the ratio between front and back performance was very similar in all three groups (Figure 5). To support the role of movement in spatial representation, we showed that typical children present a better performance in frontal spatial tasks compared to the back, while children with visual plus motor deficits perform similarly in the two spaces. The hemiplegic group with only motor impairment showed a small difference between the representation of the front and back space, with better performance in the frontal space, suggesting some kind of visual compensation. This is in agreement with Aggius-Vella’s papers [25,27,44], where it is shown that the space around the body is split up into different portions and not encoded as a unitary dimension. Importantly, these different portions of space are sensory, and motor depended: in other words, they are differently shaped by senses and the possibility to move. In particular, the authors demonstrated that visual sensory feedback and motor control lead to a more accurate representation of auditory frontal space around the chest rather than auditory frontal space around the foot. This suggests that the possibility to move significantly reduces error in localizing frontal sounds around the chest, probably because in this space, which is a restricted space around us, and reachable by our limbs (i.e., the peripersonal space), we are used to integrating sensory feedback with actions. So sound localization, at this body level, could be seen as a sort of reaching.

The visual-spatial localization task was adopted in order to investigate if and how motor impairment affects spatial skills independently of the sensory modality of presentation. Our results showed that H children performed similarly to typical subjects and HV subjects. At the same time, worse performance in the HV-group was found compared to the typical, suggesting a possible compensation of vision in an easy visual-spatial task. However, more dedicated experiments are needed in order to confirm the role of movement and vision during development in visual-spatial tasks.

## 5. Conclusions

To conclude, our results in hemiplegic children suggest that movement is important for the development of auditory spatial representation. As predicted, we demonstrated that a partial motor deficit impairs the processing of spatial location of sounds but not their content. Given this new association, these findings open new possibilities for multisensory training based on sensory-motor feedback to restore spatial representation in children with motor disorders.

## Figures and Tables

**Figure 1 children-09-01055-f001:**
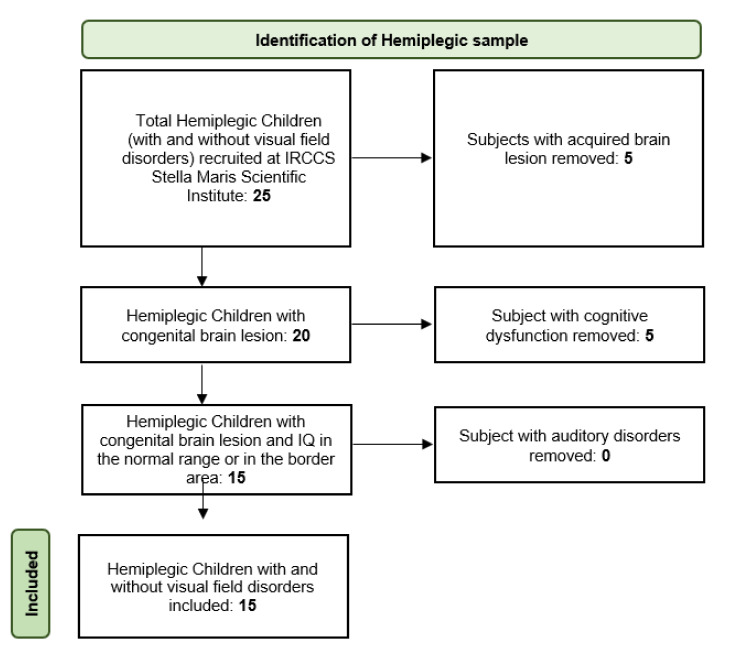
Flowchart showing the selection process for inclusion of eligible patients.

**Figure 2 children-09-01055-f002:**
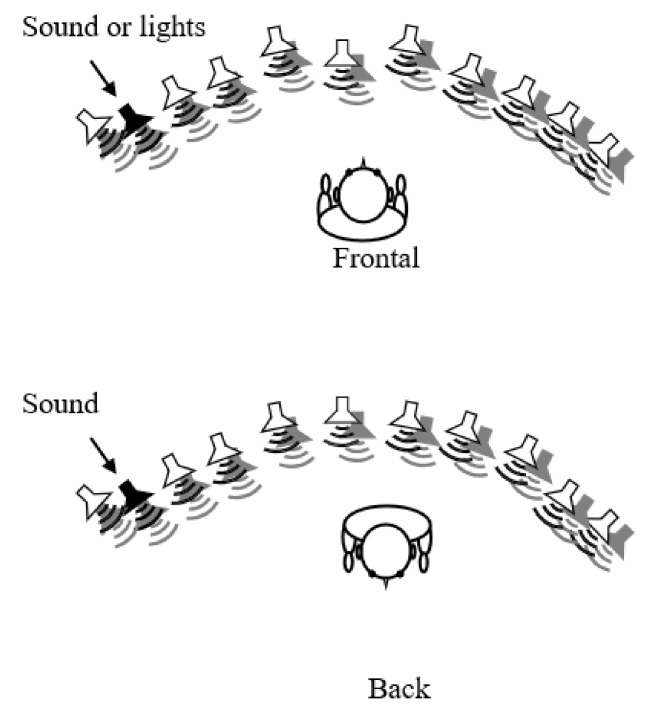
Set up schematic representation.

**Figure 3 children-09-01055-f003:**
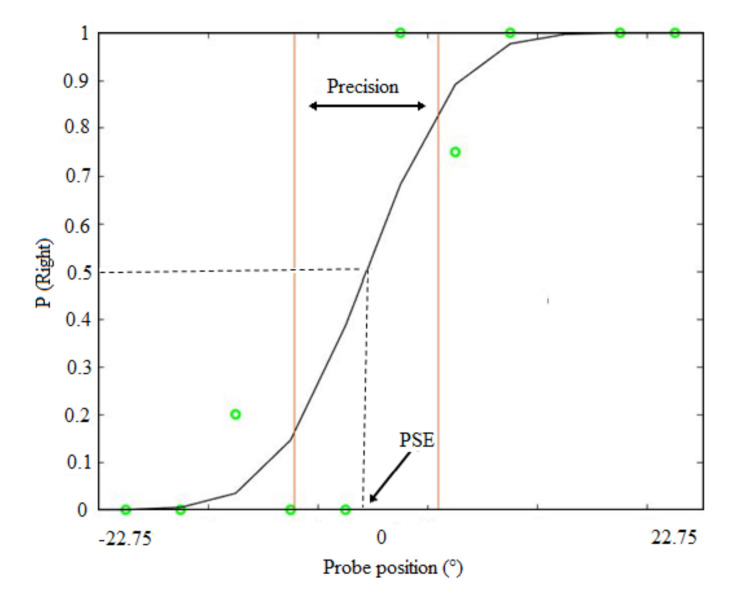
Example of a psychometric function from a typical participant for data collected in the bisection task.

**Figure 4 children-09-01055-f004:**
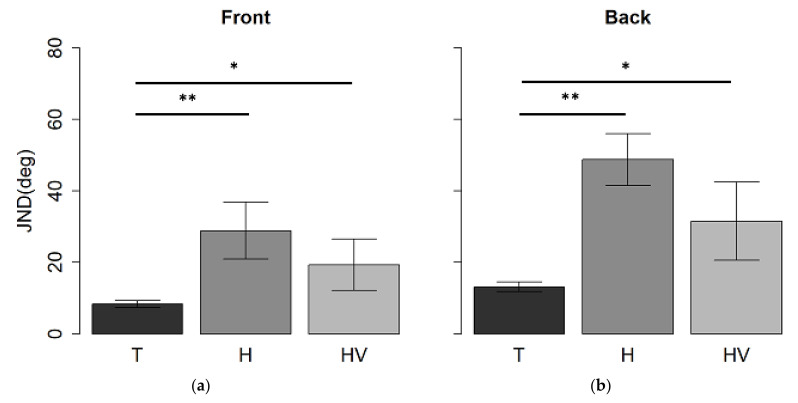
Mean values and standard deviation expressed as JND at auditory spatial discrimination task in (**a**) front presentation and (**b**) back presentation in T, H, and HV children. * *p*-value < 0.05; ** *p* < 0.01.

**Figure 5 children-09-01055-f005:**
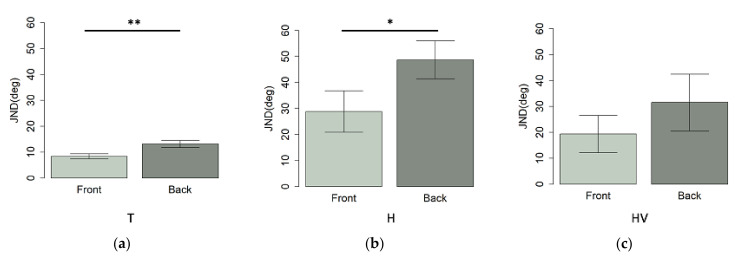
Different mean values between front and back performance in (**a**) T children, (**b**) H children and (**c**) HV children. * *p*-value < 0.05; ** *p* < 0.01.

**Figure 6 children-09-01055-f006:**
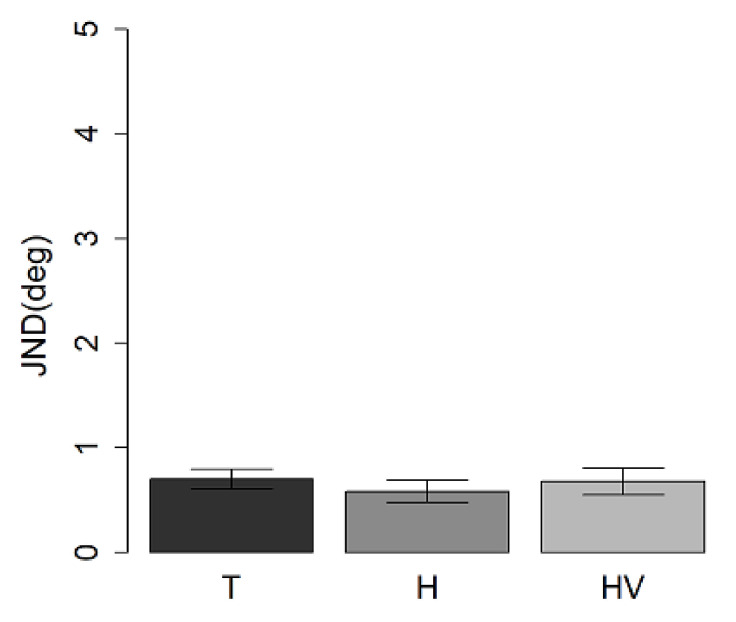
Ratio front/back at the auditory spatial task in the three different groups.

**Figure 7 children-09-01055-f007:**
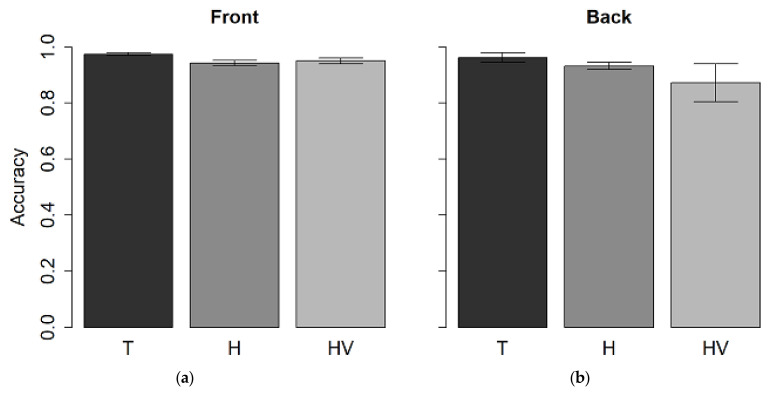
Mean values and standard deviation expressed as accuracy at pitch discrimination task in (**a**) front presentation and (**b**) back presentation in T, H, and HV children.

**Figure 8 children-09-01055-f008:**
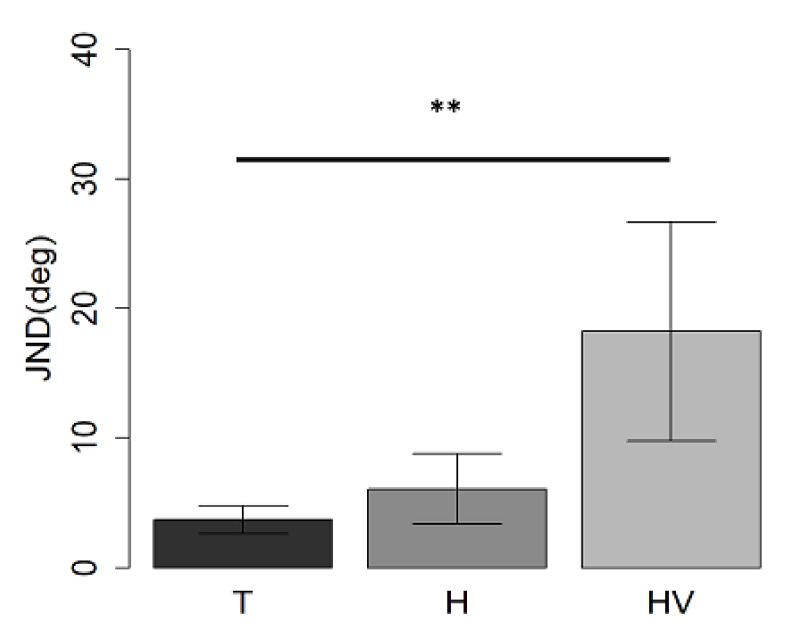
Mean values and standard deviation expressed as JND in the visual-spatial discrimination task in T, H, and HV children. ** indicates a *p*-value < 0.01.

**Table 1 children-09-01055-t001:** Description of the hemiplegic group.

			Lesion			Outcome		
Patient ID	Sex	Age (Years)	Side	Site	Type	Hemiplegia	Visual Field	MACS Scale
1	f	8	R	T,P	PVL:> involvement of the right side	L	L HH	3
2	m	11	L	F,T,P	MCA Infarction: Main branch	R	R HH	5
3	m	7	L	sc	Venous Infarction	R	No	2
4	f	8	L	P,sc	Parasagittal arterial infarction	R	No	1
5	f	11	R	sc	MCA: lenticular infarction	L	L Reduction	4
6	f	7	L	T,P,O	MCA Infarction: Main branch	R	R HH	3
7	f	8	R	P,sc	PVL	L	No	1
8	f	8	R	sc	Venous infarction	L	No	1
9	m	10	L	F,T, P,sc	MCA Infarction: Main branch	R	No	1
10	m	5	R	F	Parasagittal arterial infarction	L	No	2
11	f	5	L	F,P,sc	MCA Infarction: Main branch	R	No	3
12	m	11	L	sc	MCA: lenticular infarction	R	No	3
13	f	14	L	F,T,P,sc	MCA Infarction: Main branch	R	RQ upper	2
14	f	7	L	F,T,P,O,sc	MCA Infarction: Main branch	R	R HH	3
15	m	14	L	sc	MCA: lenticular infarction	R	No	1

Abbreviations: f = female; m = male; R = right; L = Left; PVL = periventricular leukomalacia; F = frontal; T = temporal; P = parietal; O = occipital, sc = subcortical structures; MCA = middle cerebral artery; HH = Homonymous Hemianopia; Q = quadrantanopia; MACS = Manual Ability Classification System.

## Data Availability

The data that support the findings of this study are available from the corresponding author upon reasonable request.

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
