# Peer review of "Motor Influence in Developing Auditory Spatial Cognition in Hemiplegic Children with and without Visual Field Disorder"

_children, 2022, doi:10.3390/children9071055_

Round 1

Reviewer 1 Report

The authors investigated the audio and visual spatial representation in hemiplegic children. I think these are interesting results have been obtained based on experimental tasks.

Major issue

Patients

The authors need to add the information about the inclusion criteria of the subject.

How did the authors confirm that the subjects do not have auditory disorders?

In this research plan, it is necessary to confirm that the hearing of the subject is normal and there is no difference between the left and right.

Didn't the hemiplegic subjects have contralateral auditory neglect as reported in the paper (ref 39)? If present, I think it will greatly affect the results.

 Did anyone have basal ganglia lesions and present with dystonic cerebral palsy?

 Discussion

I recommend to investigate the relationship between the MACS scale and the JND results in order to discuss the effect of fine motor ability on auditory spatial representation.

 Minor issue

On page 3, line 2 in the section 2.2 Setup and stimuli; led" is a typographical error in "LED". 

On page4, line 1; The number of hemiplegic children does not match the other descriptions.

Author Response

Major issue

Patients

The authors need to add the information about the inclusion criteria of the subject.

Thank you for you suggestion. We added mor information in this way

“Inclusion criteria for hemiplegic children were: i) congenital brain lesion; ii) intelligence quotient in the normal range or in the border area;  iii) a behavioral or computerized evaluation of the visual field, iv) no history of hearing impairment  (all children underwent an audiometric examination that was found to be normal in the first few months of life); iv) no auditory neglect and v) normal or corrected-to-normal vision and  absence of peripheral visual deficits. “

How did the authors confirm that the subjects do not have auditory disorders?

All subjects have done at least one audiometric examination in their life as we added now in the inclusion criteria

In this research plan, it is necessary to confirm that the hearing of the subject is normal and there is no difference between the left and right.

We enrolled subject only with a normal audiometric examination and without differences between left and right side

Didn't the hemiplegic subjects have contralateral auditory neglect as reported in the paper (ref 39)? If present, I think it will greatly affect the results.

In all the enrolled subjects we excluded the presence of auditory neglect (We added this in inclusion criteria).

We tested auditory neglect in all children before subjecting them to the exercises on space. This was made possible by an instrument called AVDESK that we often use for rehabilitation of hemianoptic subjects.The instrument is extremely versatile, and there is the possibility of doing auditory-only tests and recording reaction times in the perception of sounds coming from different spatial positions on both left and right.  The loudspeakers are positioned at specific eccentricities: 8°, 24°, 40°, 56°, 72°, and 90° to both left and right (for more details on the instrument you can read: Development and Implementation of a New Telerehabilitation System for Audiovisual Stimulation Training in Hemianopia buy Tinelli et al.  2017 Front. Neurology).

No omission was done and no difference was found in reaction times when sounds were on the left or on the right both in children with right brain  lesion and with left brain lesion. However we must consider that our sample is composed of only 10 children with left brain lesion and only 5 children with right brain lesion. 

This resulti is different respect to that found by Martin’s paper but: i) our sample is very small compared to the sample reported in that article, and we only have 5 subjects with left lesions, and ii) secondly, our sample has very narrow inclusion criteria. In fact we enrolled only hemiplegic children having an IQ in the normal or borderline area while this information has not been reported in Martin's  article

 Did anyone have basal ganglia lesions and present with dystonic cerebral palsy?

Only one child with dystonic cerebral palsy was enrolled and he was the only one with basal ganglia lesion

 Discussion

I recommend to investigate the relationship between the MACS scale and the JND results in order to discuss the effect of fine motor ability on auditory spatial representation.

No correlation was found between JND front sound localization and MACS’ levels (spearman p.value = 0.721. we added this paragraph in the result part:

“3.4 Correlation between MACS values and JND at front auditory spatial discrimination task

No significative correlation was found between MACS’ values and front auditory spatial discrimination task in the whole group of hemiplegici children (H+HV) (p=.721)”

 Minor issue

On page 3, line 2 in the section 2.2 Setup and stimuli; ”led" is a typographical error in "LED". 

Thank you. We changed it

On page4, line 1; The number of hemiplegic children does not match the other descriptions.

We changed the phrase in this way: “In order to understand the role of vision in basic spatial cognition, all healthy children and 13 hemiplegic children (4 with visual field disorder and 9 without visual field dis-order) performed an additional control task: the visual-spatial localization task. Two subjects refused to do also this task.”

Reviewer 2 Report

The manuscript: "Motor influence in developing auditory spatial cognition in hemiplegic children with and without visual field disorder" is well written and organized. The information is completed and give a great knowledge about the topic.

The different parts are well structured, but I suggest introduce a flowchart with the patients enrollement.

Congratulations for the work

Round 2

Reviewer 1 Report

It has been properly revised.